# Comparison of Healthiness, Labelling, and Price between Private and Branded Label Packaged Foods in New Zealand (2015–2019)

**DOI:** 10.3390/nu13082731

**Published:** 2021-08-09

**Authors:** Teresa Castro, Sally Mackay, Leanne Young, Cliona Ni Mhurchu, Grace Shaw, Essa Tawfiq, Helen Eyles

**Affiliations:** 1Department of Epidemiology and Biostatistics, The University of Auckland, Auckland 1010, New Zealand; sally.mackay@auckland.ac.nz (S.M.); shagr003@student.otago.ac.nz (G.S.); h.eyles@auckland.ac.nz (H.E.); 2Nutrition Section, The University of Auckland, Auckland 1010, New Zealand; 3National Institute for Health Innovation, The University of Auckland, Auckland 1010, New Zealand; leanne.young@auckland.ac.nz (L.Y.); c.nimhurchu@auckland.ac.nz (C.N.M.); e.tawfiq@auckland.ac.nz (E.T.); 4The George Institute for Global Health, Newtown, NSW 2042, Australia

**Keywords:** supermarket packaged foods, private labels, generic labels, branded labels, health star rating, sugar, sodium, healthiness, price, public health policy

## Abstract

We aimed to compare New Zealand private label (PL) and branded label (BL) packaged food products in relation to their current (2019) healthiness (sodium and sugar contents, and estimated Health Star Rating (HSR) score), display of the voluntary HSR nutrition label on the package, and price. Healthiness and HSR display of products were also explored over time (2015 to 2019). Data were obtained from Nutritrack, a brand-specific food composition database. Means and proportions were compared using Student *t*-tests and Pearson chi-square tests, respectively. Changes over time were assessed using linear regression and chi-square tests for trends (Mantel–Haenzel tests). Altogether, 4266 PL and 19,318 BL products across 21 food categories were included. Overall, PL products in 2019 had a significantly lower mean sodium content and price, a higher proportion of products with estimated HSR ≥ 3.5/5 (48.9% vs. 38.5%) and were more likely to display the HSR on the pack compared with BL products (92.4% vs. 17.2%, respectively). However, for most food categories, no significant difference was found in mean sodium or sugar content between PL and BL products. In the period 2015–2019, there were no consistent changes in estimated HSR score, sodium or sugar contents of PL or BL products, but there was an increase in the proportion of both PL and BL products displaying HSR labels. In most food categories, there were PL options available which were similar in nutritional composition, more likely to be labelled with the HSR, and lower in cost than their branded counterparts.

## 1. Introduction

New Zealand (NZ) has a high prevalence of nutrition-related disease (NCD) [1], with poor diets characterized by energy-dense, nutrient-poor foods and beverages, accounting for nearly 20% of illness and early death in 2017 [1]. Furthermore, there is an inequitable food environment in NZ which promotes the consumption of unhealthy foods [2,3,4,5,6,7], including the steady growth of the packaged food industry, which consequently has an important role to play in improving population diets and preventing NCDs [2]. In NZ, supermarkets account for ~75% of all purchases of packaged foods [8], and the supermarket food environment consists of a duopoly of two supermarket retailers: Foodstuffs and Woolworths [9]. These two supermarket retailers provide groceries to eight supermarket chains across the country [9].

The availability of private label (PL) and branded label (BL) products in supermarkets is important for generating price competitiveness and to offer consumers options in terms of quality and variety [10]. Both supermarket retailers in NZ offer PL options, also known as “own-brands”, “generic brands”, “store brands”, or “economy lines”. In 2020, PL sales accounted for 10.2% of packaged foods sold in NZ stores and online [8]. In 2020, Foodstuffs reported that they were on target to reach $1.3 billion in sales from PL products [11]. In 2019, Woolworths Group expanded their PL range and released 640 new PL products into the market [12]. Given the relevant presence of PL products in NZ supermarkets, it is important that these products are equitable from a health perspective.

Since 2016, both NZ supermarket retailers have made commitments to improve the healthiness of their PL products, specifically through reformulation focused on reducing sugar, sodium, and saturated fat content [13,14,15]. For example, one retailer has committed to all PL products being nutritionally on par, or better than, the average comparable BL products (by 2018) [13]. Both retailers have also committed to displaying the voluntary Health Star Rating (HSR) nutrition label on all applicable PL products [13,14,15]. The HSR system was introduced into New Zealand and Australia in mid-2014 as a front-of-pack labelling scheme endorsed by the government to enable consumers to easily compare the healthiness of similar types of products on a scale of 0.5 to 5.0, with a higher score indicating a healthier product [16]. Products with an HSR ≥ 3.5 stars are generally considered a healthier choice [17].

Two previous NZ studies have examined differences in the healthiness of PL and BL products, with both examining sodium content [18,19], and one examining price [18]. In 2002, across 11 of 15 food categories assessed, mean sodium content was found to be lower for most PL compared with BL options in analyses involving unmatched products. This same study also found that, for 11 food categories, PL products were, on average, cheaper than BL options. This finding contrasts with that described by Monro et al. (2015) [19], where the mean sodium content of PL products was found to be higher than that of BL products. However, it is important to highlight the small number of food categories (n = 8) and variation in product types included in this latter study, which limits the generalizability of the findings. 

To the best of our knowledge, there are no recent and comprehensive studies in NZ comparing the healthiness and price of PL and BL packaged food products, or changes in their healthiness over time. Consumers have the right to be informed when looking for healthier, cost-effective food options, and this is particularly important for addressing equity. Furthermore, NZ food retailers need to know how healthy their PL products are compared with branded options. Therefore, our aim was to compare the healthiness, display of HSR, and price of PL and BL packaged food products sold at major NZ supermarket chains. Specific questions were: (1) Do the healthiness, display of HSR and price differ between PL and BL food products on the market in 2019? (2) Has the healthiness and display of HSR on PL and BL packaged food products changed over time (five years from 2015 to 2019)?

## 2. Materials and Methods

### 2.1. Outcomes and Data Sources

In this study, the following indicators were described and compared between PL and BL packaged food categories (FCs):(i)Healthiness: Data on the sodium and sugar contents (mg/100 g) and estimated Health Star Rating (HSR) were extracted from the Nutritrack database for the years 2015 to 2019. Nutritrack is a packaged food database managed by the National Institute for Health Innovation at the University of Auckland and includes information for packaged foods sold in four major NZ supermarket chains (New World, Four Square, Countdown and PAK’nSAVE). Annual cross-sectional supermarket surveys are undertaken using a systematic process at the same time each year (February to May) in Auckland, New Zealand. Photographs of packaged foods and beverages that display a nutrition information panel (NIP) are taken using a customized smartphone application and names, brands, labelling, ingredients and NIP information are entered into a secure online system [20]. Information is collected for ~75% of unique packaged foods and beverages purchased in NZ [8]. By 2018, only 21% of the NZ supermarket packaged products displayed the manufacturer-calculated HSR score on the pack [2]. Thus, for the purposes of this study, we estimated the HSR score for all products using the stepwise approach and the HSR Calculator 2018 provided by The New Zealand Ministry for Primary Industries; further details are available in Tawfiq et al. (2021) [5]. Estimated HSR scores were categorized as <3.5 stars (unhealthy) and ≥3.5 stars (healthy) for analyses [16].(ii)Products displaying HSR on the pack: Information on whether products were displaying HSR on the pack was also sourced from the Nutritrack database for the years 2015 to 2019. This outcome was classified as “yes” or “no”.(iii)Price: Mean price for each product in Nutritrack 2019 (mean NZ$/product package) was calculated using price information from the Nielsen New Zealand Homescan^®^ panel between October 2018 and October 2019. Nielsen market research data are one of the largest and most up-to-date datasets available to monitor household food purchases [21]. The Nielsen New Zealand Homescan^®^ panel is a sample of approximately 2500 households, designed to be representative of NZ households in terms of geographic and demographic characteristics. Nielsen New Zealand Homescan^®^ excludes data for households who scan items inconsistently or do not meet the minimum spending criteria. We used data for 1,800 NZ households who purchased food in stores, and this approach was consistent with that used for in-store data collection for Nutritrack 2019. Households are based in major and secondary urban sites (according to the definition of Statistics NZ [22]), which accounts for 92% of the country’s population [5]. Price information was estimated from all product purchases made by panel members between October 2018 and October 2019. We excluded all pricing data from stores other than supermarkets, grocery stores, fruit and vegetable stores, convenience stores, fish and meat stores, and bakeries. Using the product barcodes as the key linking variable, information on the mean price of each product (NZ$) in NZ Nielsen Homescan^®^ was linked to Nutritrack. After data merging, the mean price of product packs was converted to mean price (in NZ$) per 100 g of product, to enable comparison across different package sizes.

### 2.2. Selection of Products, Exclusion Criteria and Data Preparation

In Nutritrack, individual products are categorized into a standardized hierarchical structure of five levels (L1 to L5), the top three comprising 15 food groups, 59 categories and 177 smaller subcategories [23,24]. Information on PL or BL status was retrieved from company websites and manually added to each unique product in the Nutritrack data [20]. The selection of the food categories in Nutritrack for inclusion in the current analyses was first based on the rationale that reformulation of products within that category should be feasible, e.g., fresh dairy milk was excluded. In order to guarantee statistical power for the analyses, food categories also had to have at least 30 [25] PL products available in 2019 for the comparison of means, or at least 100 PL products available across all years from 2015 to 2019 for comparisons of mean changes over time. Food categories were initially selected using Nutritrack food group classification level 2 (L2), for example, fish/processed fish. However, when a food category at L2 contained a range of nutritionally diverse products, minor food categories at Nutritrack levels (L3, 4 and 5) were selected instead (for example, canned fish was used rather than the aggregated group fish and seafood products/processed fish). In total, 21 FCs were selected for inclusion, comprising 24,205 products in the period 2015–2019. If selected food categories at any level contained any minor food categories at L3, L4 or L5 with less than five PL products in the period 2015–2019, products within the minor (s) category at L3, L4 or L5 were excluded (n = 408; 1.7%). For example, anchovies were not included in the canned fish category because there were <5 anchovy products across the five years. Products where nutrient data were only available in reconstituted form, and products with multiple NIPs, such as meal kits, were also excluded (n = 209; 0.9% and n = 4; 0.02%, respectively). Thus, the total number of products included in the analyses from 2015 to 2019 was 23,584 (4266 PL and 19,318 BL products). Products included in analyses corresponded to 31.1%, 40.7% and 29.9% of all products, PL products, and BL products available in the Nutritrack database, respectively (2015–2019). Appendix A presents the selected food categories, their minor food categories and the number of products assessed in each category (in total, for PL and for BL).

Sugar content of PL and BL products was not compared within food categories that are not key sources of sugar, i.e., canned fish, canned vegetables, pickled vegetables, salted nuts, processed meats, and crispy and salty snacks. Similarly, sodium content was not compared for ice-cream and fruit in syrup/juice, as these are not major sources of sodium. In total, 170 (0.7%) products had missing information for sodium and sugar, so it was not possible to estimate and HSR for them. Estimations were not calculated for a further 152 (0.6%) products as there were errors in their sodium and/or sugar contents in Nutritrack. Appendix A shows the number and percentage of products with missing information for sugar content, sodium content, or estimated HSR, according to the food category. Information was available for all products and years on whether HSR was displayed on pack. Of the 4896 selected products in Nutritrack in 2019 (PL and BL), 431 (8.8%) were not included in the Nielsen Homescan data for 2019, and, thus, information on price for these products was missing.

### 2.3. Statistical Analyses

Descriptive statistics were performed to describe means and standard deviations (SD), value ranges and proportions. There were not enough PL products available to allow paired comparisons to assess product reformulation over time. Therefore, in this study, we compared means and proportions in 2019 and changes in means and proportions in the period 2015–2019 (overall and at the food category level).

Food categories with 30 or more products were considered sufficiently large for the central limit theorem to apply [25]. T-tests for independent samples were applied to compare statistically significant differences in means between PL and BL products in 2019. Pearson chi-square tests were performed to assess whether there were statistically significant differences in proportions of PL and BL products displaying HSR on the pack and with estimated HSR ≥ 3.5 in 2019. 

Mean changes in sugar and sodium content in the period 2015–2019 were assessed separately within BL and PL products (overall and by FC). To estimate the average change in sodium (mg/100 g) or sugar (g/100 g) contents from 2015 to 2019, linear regression models were performed with sodium or sugar as the dependent variable. Year was included in the model as the independent variable—as a continuous variable, coded as: 2015 = 0, 2016 = 0.25, 2017 = 0.50, 2018 = 0.75, and 2019 = 1 [3,26]. Mean percentage change in sugar and sodium content across the five years was calculated by dividing the adjusted mean change in sugar or sodium from 2015 to 2019 by the mean value in 2015 (multiplied by 100%). Overall and within the food categories, five-year trends in the proportion of PL and BL products with HSR ≥ 3.5, and products displaying the HSR were examined using chi-square tests for trends (linear-by-linear associations using Mantel–Haenzel tests). Analyses of changes over time were performed for all PL and BL food categories as all contained at least 100 products with information available in the period 2015–2019. Analyses were performed using SPSS software (version 25, IBM SPSS Statistics), and all tests were two sided at the level of significance of 5%.

## 3. Results

### 3.1. Comparison of PL and BL Products in 2019

#### 3.1.1. Healthiness: Mean Sugar and Sodium Content and Proportion of Products with Estimated HSR ≥ 3.5 

Table 1 and Table 2 describe and compare the indicators of healthiness between PL and BL products by food category in 2019. Overall, PL products had statistically significantly lower mean sodium content than BL products. However, there were significant differences in mean sodium content between PL and BL products for only two of the 19 food categories assessed (canned fish and canned vegetables, both with lower means for PL products). Overall, there were no differences in the mean sugar content of PL and BL products, and a significant difference in mean sugar content between PL and BL products for only one of the 14 food categories assessed, i.e., canned fruit, with a lower mean for PLs (Table 1). 

Overall, there was a statistically significantly higher proportion of PL products with an estimated HSR ≥ 3.5 (48.9%) compared to BL products (38.5%; Table 2, 2019 column). However, there were statistically significant differences in the proportion of products with an estimated HSR ≥ 3.5 between PL and BL for just three of the 21 food categories assessed, i.e., PL canned fruit, and savoury spread and dips had a higher proportion of products with HSR ≥ 3.5. Cereal bars had a lower proportion of products with HSR ≥ 3.5 (Table 2). In 2019, the proportion of PL products with an estimated HSR ≥ 3.5 ranged from 0% (for sweet biscuits, cakes/muffins, ice-cream and mayonnaise/salad dressings) to 100% (for everyday sliced breads, canned fish, canned fruit and peanut butter and other nut-based spreads). Within BL products, the proportion of products with estimated HSR ≥ 3.5 ranged from 0% (for cakes/muffins; mayonnaise/salad dressings) to 98.9% (for canned vegetables) (Table 2).

#### 3.1.2. Display of HSR on the Pack

The proportions of PL and BL products (overall and for each food category) displaying HSR on the pack in 2019 are shown in Table 3. Overall, PL products had a substantially higher prevalence of HSR label display than BL products (92.4% vs. 17.2%). Within food categories, more PL products displayed HSR on the pack compared to BL counterparts (with the only exception being cereal bars, where there was no difference in HSR uptake between PL and BL products in 2019). The proportion of PL products with HSR displayed on the pack ranged from 41.2% (cereal bars) to 100% (everyday sliced breads and peanut butter and other nut-based spreads). Within BL food categories the proportions ranged from 1.0% (cakes/muffins) to 55.7% (breakfast cereals; Table 3).

#### 3.1.3. Price

In 2019, overall, the mean price of PL products was statistically significantly lower than the mean price of BL products. There were also statistically significant differences in mean price between PL and BL products for 11 of the 16 food categories where it was possible to compare price. Mean prices of savoury biscuits, sweet biscuits, other breads, breakfast cereals, cereal bars, canned fish, canned fruit, canned vegetables, savoury spreads and dips, peanut butter and other nut-based spreads and crisps and salty snacks were significantly lower for PL options than BL options. There were no significant differences in mean price between PL and BL products for the other food categories assessed (Table 4).

### 3.2. Changes in Healthiness and HSR Display from 2015 to 2019 

#### 3.2.1. Healthiness Changes in Mean Sodium Content

Information on the mean sodium content across the five years for all products and for PL and BL products separately, as well as in their minimum and maximum values, is available in Appendix A. Figure 1 shows the mean change in sodium content from 2015 to 2019. 

Overall, there were no significant changes in mean sodium content within all PL products and there was a significant mean sodium [mg/100 g (95% CI)] reduction of −37.8 (−57.8; −17.8) within all BL products (average percentage change of −7.5%). At the food category level, three PL food categories significantly reduced mean sodium content over time (everyday sliced breads, other breads, and cakes/muffins), with the mean percentage change >10% and the respective mean [g/100 g (95% CI)] reductions of −61.4 (−86.3; −36.4); −94.5 (−142.4; −46.7) and −109.8 (−190.5; −29.0). The mean sodium content of PL sweet biscuits increased over time by 58.1 mg/100 g (95%CI: 18.6; 97.7), corresponding to a percentage increase of 28.3% (Figure 1A). Similarly, three BL food categories significantly reduced mean sodium content over time (savoury biscuits, salamis, hams and bacon and crisps and salty snacks), all with an average percentage reduction of <10% and with the respective mean (95% CI) reductions of: −69.7 (−114.7; −24.8); −82.9 (−151.9; −13.8) and −59.2 (−116.6; −1.7) mg/100 g (Figure 1B).

#### 3.2.2. Healthiness: Changes in Mean Sugar Content

Information on the mean sugar content across the five years for all products and for PL and BL products separately, as well as in their value ranges, is available in Appendix A. Figure 2 shows the change in sugar content from 2015 to 2019.

Overall, there were no significant changes in mean sugar content within all BL products and there was a significant mean sugar [g/100 g (95% CI)] reduction of −1.87 (−3.3; −0.04) within all PL products (average percentage change of −11.2%). At the food category level, two PL categories significantly reduced sugar content over time, i.e., canned fruit, and peanut butter and other nut-based spreads reduced mean sugar content by, respectively [g/100 g (95%CI)]: −2.02 (−3.20; −0.84) and −5.94 (−10.68; −1.20), corresponding to an average percentage drop of >10%. The mean sugar content of everyday sliced breads increased over time by 1.21 g/100 g (95%CI: 0.60; 1.83), (Figure 2A). Four BL food categories significantly reduced sugar content over time, i.e., breakfast cereals and pasta sauces reduced mean sugar content by [g/100 g (95%CI)]: −1.31 (−2.60; −0.04) and −5.26 (−9.01; −1.44), respectively, corresponding to an average percentage drop of >10%. Branded label cereal bars and ice-creams showed a mean sugar reduction of [g/100 g (95%CI)]: −2.49 (−4.10; −0.89) and −1.67 (−2.44; −0.91), respectively, corresponding to an average percentage drop of <10%. Across the five years, BL savoury biscuits increased mean sugar content by 0.59 g/100 g (0.06; 1.10) (Figure 2B). 

#### 3.2.3. Healthiness: Changes in Proportions of Products with an Estimated HSR ≥ 3.5

Table 2 describes the changes in the proportion of PL and BL products with an HSR ≥ 3.5 in the period 2015–2019 (overall and by food category). Across the five years, overall, there were no significant changes in the proportion of PL or BL products with an HSR ≥ 3.5. However, analyses within food categories indicated statistically significant increases in the proportion of products with estimated HSR ≥ 3.5 over time for four PL food categories (everyday sliced breads, other breads, breakfast cereals and spreads II) and four BL food categories (spreads I, crisps and salty snacks, cereal bars and ice-creams) (Table 2).

#### 3.2.4. Changes in the Proportion of Products Displaying HSR

There was a statistically significant increase in the proportion of products displaying HSR over time within all PL and BL food categories, the only exception being BL cakes/muffins and BL mayonnaise and salad dressings (Table 3).

## 4. Discussion

### 4.1. Statement of Principal Findings

In 2019, PL products had, overall, a lower mean sodium content in relation to all BL products. Overall, PL and BL products had similar mean sugar contents, and the mean sodium and sugar content of most food categories was not significantly different between PL and BL products. Overall, a higher proportion of PL products had an estimated HSR ≥ 3.5 (48.9%), compared to BL products (38.5%). Considerably more PL products displayed the HSR on the pack than BL products (92.4% vs. 17.2%), and PL products were overall lower in price than BL options. There were no consistent changes over time (2015–2019) in any of the healthiness outcomes (sodium, sugar, or estimated HSR) of PL and BL products, but an increase in display of HSR on the pack was observed over time for all PL and BL food categories. 

### 4.2. Findings in Relation to Other Studies 

#### 4.2.1. Healthiness

Results of our study showing a lower mean sodium content of PL products overall differ to those reported by a previous NZ study that compared the sodium content of PL and BL products in supermarkets between 2003 and 2013 [19]. Note, however, that the previous study compared matched means of PL and BL products available in both years and in only eight categories, rather than comparing means of PL and BL overall [19]. These aspects limit direct comparisons to our findings. Results of our study, however, align with studies conducted in Australia [27,28,29] and other countries [30,31,32,33], which showed that, despite differences in healthiness for a small number of FCs between PL and BL products (in both directions), overall, there were no systematic differences in healthiness between PL and BL products [27,28,29,30,31,32,33]. These studies used various methods. Ahuja et al. (2017) [30] undertook chemical analysis of 1,706 samples of PL and national brand products between 2010 and 2014 in the United States (US). In 2010 and 2012, a study in the United Kingdom assessed and compared the nutritional quality of 32 own brands and market brands processed foods most frequently consumed in the country. Products were sourced from supermarkets and their nutritional quality scoring was calculated according to the Food Standards Agency’s Traffic Light System [31]. A Swiss study compared the nutritional quality of over 4000 processed foods distributed across 26 food categories. No differences were found between PL and BL products for total energy, protein, fat, and carbohydrates for most food categories. However, PL products had a lower fat, saturated fatty acid, and sodium content [33] in some food categories. In Australia in 2017, a study conducted in four major supermarket chains assessed 6269 products and found no differences in mean HSR in matched comparisons of PL and BL for any of the 10 food categories assessed [27]. Another Australian study also conducted in four major supermarket chains (in Sydney) but assessing a larger number of products (15,680 products, distributed in 15 food categories) found in 2013 that new supermarket PL products were 11% lower in sodium in relation to their BL counterparts [28]. An older study (2006–2008) involving 10 Australian supermarkets and 3204 products from 15 food categories identified that the contents of total and saturated fat were significantly greater for five and seven PL food categories, respectively, in relation to BL options. For sodium content, there were significant differences between PL and BL for seven food categories, but with no consistency in direction [29].

#### 4.2.2. Display of HSR

Concern has been expressed by public health experts that voluntary uptake of the HSR label is slow and therefore it should be made mandatory [33]. Front-of-pack labelling provides visual information on product nutritional contents and studies have shown that it influences consumer’s knowledge [34,35] and products reformulation [34]. Recent systematic review and meta-analyses including controlled experimental/intervention and interrupted time series found that findings about influence of front-of-pack labelling on consumers’ consumption were limited and inconsistent. However, evidence from experimental and ‘real-life’ studies shows that front-of-pack labelling encouraged healthier purchasing [35]. An online randomized-controlled study of a large representative British sample found that front-of-pack labelling improved participants‘ ability to correctly rank products according to their healthiness [36]. A non-experimental prospective study reported that food reformulation occurred after the first phase of the Chilean Food Labelling and Advertising Law, with significant decreases in the amount of sugars and sodium in several groups of packaged foods and beverages between 2015 and 2017 [34]. 

A previous NZ study describing the state of the packaged food supply in 2018 indicated that products (PL and BL aggregated) displaying the HSR on the package had a higher mean HSR than products not displaying HSR values (mean ± SD, 3.2 ± 1.3 versus 2.5 ± 1.4, *p* = 0.000) [2]. Among the products examined in the period 2015–2019, our study indicated much greater uptake of HSR by PL products (92.4% in 2019) than BL products (17.2% in 2019). An Australian study in 2017 also reported a significantly higher proportion of supermarket PL products displaying HSR (57%) than BL products (28%) [27]. 

#### 4.2.3. Price

The lower cost of PLs in relation to BLs reported in the current study corroborates with the 2003 NZ study that found lower mean price for 11 of 15 supermarket PL food categories examined (in relation to BL) [18]. A study looking at the cost of healthy and usual diets in NZ in 2015 found considerable savings (5.5%) if households purchased PL versions of brands compared to branded items [37,38]. Findings of our study are also similar to those reported in several other countries internationally, where, overall, supermarket PL products were lower priced in relation to BL options [31,32,33,39,40].

### 4.3. Findings in Relation to the Commitments Made by NZ Supermarkets

In our study, we did not evaluate separately how each of the two supermarket retailers met their commitments made in 2016, because there were insufficient PL products for most food categories assessed to provide robust comparisons. Thus, comparisons made include PL FCs of both NZ supermarket retailers combined. As previously described, both NZ supermarket retailers committed to displaying HSR on almost all PL products by 2018–2020 [13,14,15]. The findings of our study confirm that this commitment is, overall, on track, as among the food categories examined, the majority of PL products (92.4%) were displaying HSR on the package in 2019. However, further effort to increase HSR uptake in some PL food categories is still required, e.g., cereal bars, sausages and hotdogs and raw or frozen meats with flavour/coating.

Both supermarket retailers committed to improving the nutrition of their PL products. Our study found that in 2019, most PL food categories were of a similar nutritional quality to BL categories. In 2019, overall, a higher proportion of PL products had an HSR ≥ 3.5 in relation BL products (43.5% vs. 38.5%), and for 10 PL food categories the proportion of products with an HSR ≥ 3.5 was >50%. We found from 2015 to 2019 that only three PL food categories and two PL food categories changed, respectively, the mean sodium and sugar contents (with average reduction > 10%). Together, these findings indicate that the commitments of supermarkets retailers have been partially met, but more work is needed to increase the proportion of products with HSR ≥ 3.5 across all types of foods.

There are no public commitments made by the NZ supermarket retailers on price of PL products [11,13,14,15]. We believe that this is probably because PL products are usually considered lower cost options than branded products, and price is generally considered commercially sensitive. In our study, overall, PL products had a lower mean price than BL products, which indicates that, on average, PL products represent better value for money than BL options.

### 4.4. Strengths and Limitations of This Study

The strengths of our study include the fact that we assessed information on a large number of packaged foods and included data over five years to assess changes in healthiness and display of HSR over time. In total, 1/3 of packaged foods in the Nutritrack database from 2015 to 2019 were included in the analyses, corresponding to 40.7% of the PL products and 29.9% of the BL products. We also compared similar types of products and to improve the robustness of analyses, mean sodium and sugar contents were compared only for food categories that contained at least 30 PL products. Similarly, we only assessed mean changes in sodium and sugar contents over time if there were at least 100 PL products available for that period. Another strength is that, rather than using price information at a single point in time in 2019, we used information on product price over the whole year and calculated the mean price that consumers paid for products within this timeframe.

Findings of this study need to be interpreted taking limitations into account. A relevant limitation is that there were not enough PL products available within the food categories to allow for paired comparisons and to assess reformulation of individual products over time. Finally, the fact that the results of our study were not sales weighted or informed by product sales data represents another limitation. Sales data could provide valuable information on the most commonly purchased products and foods to better assess the public health impact of findings, including by sociodemographic group.

### 4.5. Implications of the Findings

In summary, PL products in major NZ supermarkets can be a good choice for consumers as they are usually lower in price, nutritionally similar to BL products, and more likely to display a HSR score. Retailers have made progress on their nutritional and labeling commitments regarding PL products. However, further positive movements can be made, including displaying the HSR on all products and establishing a systematic PL reformulation programme operating across all foods, but with an emphasis on categories with a high sales volume. These recommendations are important for public health given that PL products are driving the growth of sales in NZ supermarkets, and most NZ shoppers believe these supermarket own brands are ‘just as good or better than’ their branded counterparts [41]. 

To set a level playing field for all companies and retailers, and to help consumers make healthier choices, the government should make display of the HSR mandatory. While this study did not assess reformulation, providing targets for reformulation of common products would provide benchmarks for retailers and the wider food industry.

## Figures and Tables

**Figure 1 nutrients-13-02731-f001:**
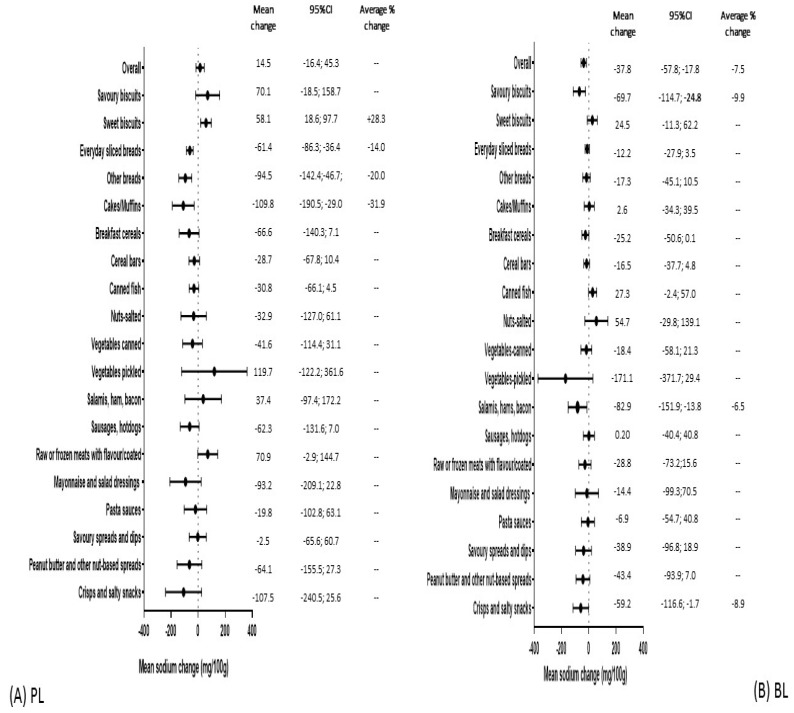
Mean sodium changes in the period 2015–2019 according to brand (overall and by food categories): supermarket private labels (**A**) and branded labels (**B**).

**Figure 2 nutrients-13-02731-f002:**
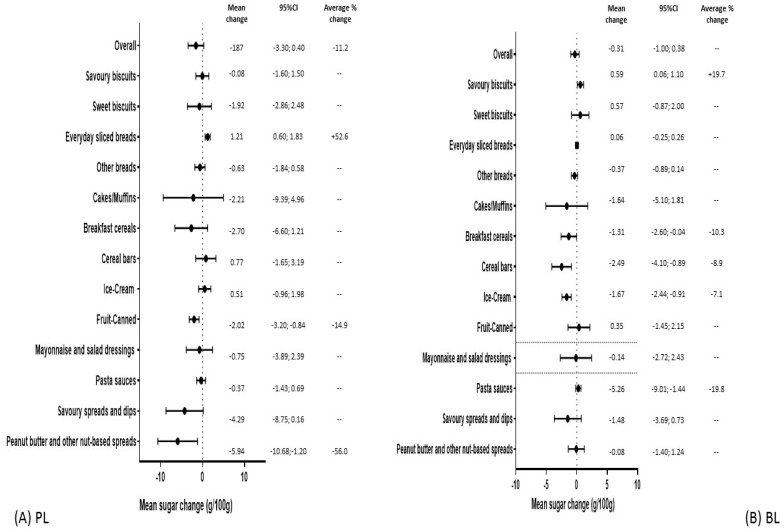
Mean sugar changes in the period 2015–2019 according to brand (overall and by food categories): supermarket private labels (**A**) and branded labels (**B**).

**Table 1 nutrients-13-02731-t001:** Mean (standard deviation) of sodium and sugar content of branded and private label supermarket products in 2019, by food category and overall.

Food Categories	Sodium (mg/100 g)		*p* Value *	Sugar (g/100 g)		*p* Value *
	N	Mean (SD)		n	Mean (SD)	
Savoury biscuits			0.132			0.512
BL	295	624.6 (263.7)		295	3.0 (2.5)	
PL	50	565.7 (200.1)		50	3.4 (4.3)	
Sweet biscuits			0.154			0.876
BL	356	286.4 (135.7)		356	32.9 (11.2)	
PL	73	261.2 (143.1)		73	33.1 (9.6)	
Everyday sliced breads			^†^			^†^
BL	105	398.3 (73.3)		105	2.8 (1.0)	
PL	9	373.3 (22.9)		9	3.2 (0.76)	
Other breads			0.100			0.625
BL	223	421.0 (144.9)		225	2.8 (2.0)	
PL	31	375.0 (146.0)		31	2.6 (1.5)	
Cakes/Muffins: ready-to-eat			^†^			^†^
BL	98	288.9 (138.9)		98	30.5 (12.3)	
PL	15	216.2 (174.1)		15	31.6 (13.9)	
Breakfast cereals: ready-to-eat			0.690			0.764
BL	280	165.9 (155.1)		275	17.2 (8.1)	
PL	41	176.5 (180.3)		41	17.6 (8.3)	
Cereal bars			0.212			0.357
BL	156	152.0 (106.0)		156	25.4 (9.7)	
PL	34	127.3 (95.5)		34	27.0 (4.9)	
Ice-cream			--			0.337
BL	--	--		340	22.0 (6.1)	
PL	--	--		25	23.1 (3.3)	
Canned fish			<0.001			--
BL	149	423.5 (153.8)		--	--	
PL	41	331.3 (96.0)		--	--	
Fruit—canned in syrup/juice			--			0.027
BL	--	--		71	13.4 (5.5)	
PL	--	--		67	11.7 (3.7)	
Nuts—salted			0.183			--
BL	73	459.6 (324.2)		--	--	
PL	31	375.6 (195.7)		--	--	
Vegetables—canned			<0.001			--
BL	181	221.9 (161.2)		--	--	
PL	60	135.3 (114.0)		--	--	
Vegetables—pickled			0.946			--
BL	177	1057.9 (1023.2)		--	--	
PL	29	1044.9 (484.8)		--	--	
Salamis, hams, bacon			0.794			--
BL	263	1204.3 (438.8)		--	--	
PL	44	1185.9 (392.1)		--	--	
Sausages, hotdogs			^†^			--
BL	102	764.8 (187.0)		--	--	
PL	20	616.0 (93.0)		--	--	
Raw or frozen meats with flavour/coated			0.232			--
BL	153	495.3 (204.3)		--	--	
PL	38	451.1 (200.4)		--	--	
Mayonnaise and salad dressing			0.963			0.441
BL	156	686.9 (452.4)		154	11.8(12.9)	
PL	34	690.6 (282.5)		34	10.0 (6.4)	
Pasta sauces			0.388			0.850
BL	156	407.9 (242.7)		151	4.4 (2.1)	
PL	39	371.9 (184.1)		39	4.9 (2.1)	
Savoury spreads and dips			0.526			0.196
BL	305	492.7 (344.8)		302	12.3 (14.2)	
PL	32	453.6 (157.6)		32	9.0 (9.0)	
Peanut butter and other nut-based spreads			^†^			0.662
BL	80	163.8 (151.9)		80	5.9 (3.1)	
PL	25	208.4 (160.7)		25	5.6 (3.1)	
Crisps and salty snacks			0.952			--
BL	216	625.1 (365.4)		--	--	
PL	35	621.2 (306.5)		--	--	
Overall (all products)			0.001			0.404
BL	3524	506.3 (540.5)		2608	15.5 (13.7)	
PL	681	443.3 (362.1)		475	14.9 (12.7)	

SD: standard deviation; BL: branded label; PL: private label. * *p*-values of Student *t*-tests for comparison of means of two independent samples. ^†^: Comparisons between means were not performed when PL had n < 30 products with information on sodium or sugar contents. --Nutrient content not assessed as food category does not represent relevant source of the nutrient.

**Table 2 nutrients-13-02731-t002:** Number and proportion of branded and private label products with an estimated HSR of > 3.5 by year (2015–2019), food category and overall.

Food Categories	Estimated HSR ≥ 3.5										*p* for Trend ^†^ Changes in Proportions In the Period 2015–2019
	2015		2016		2017		2018		2019		
	n	%	n	%	N	%	n	%	n	%	
Savoury biscuits											
BL	50	19.7	43	17.4	47	18.7	57	20.3	57	19.3	0.776
PL	5	10.6	11	20.8	10	16.4	6	9.5	7	14.0	0.676
Sweet biscuits											
BL	2	0.7	2	0.6	2	0.7	3	0.8	4	1.1	0.471
PL	0	0	0	0	0	0	0	0	0	0	^‡^
Everyday sliced breads											
BL	100	89.3	98	95.1	103	96.3	104	96.3	100	95.2	0.055
PL	26	81.3	24	80.0	15	100	16	100	9	100	0.011
Other breads											
BL	119	61.3	124	60.8	123	62.4	141	65.9	147	66.2	0.156
PLPL	28	45.2	32	42.1	28	59.6	22	59.5	23	74.2	0.002
Cakes/Muffins: ready-to-eat											
BL	1	1.4	0	0	0	0	0	0	0	0	0.130
PL	0	0	0	0	0	0	0	0	0	0	^‡^
Breakfast cereals: ready-to-eat											
BL	140	64.5	168	64.6	163	66.0	166	64.3	171	62.4	0.613
PL	23	52.3	30	56.6	27	58.7	23	65.7	30	73.2	0.033
Cereal bars											
BL	11	7.4	24	13.1	32	17.7	38	21.1	38	24.4 *	<0.001
PL	3	8.6	3	6.7	4	10.0	3	8.6	3	8.8 *	0.847
Ice-cream											
BL	3	1.2	5	1.8	6	2.2	7	2.2	17	5.0	0.004
PL	0	0	0	0	0	0	0	0	0	0	^‡^
Canned fish											
BL	157	96.3	127	93.4	137	92.6	130	90.3	141	94.6	0.285
PL	72	97.3	79	98.8	76	98.7	50	100	41	100	0.147
Fruit—canned in syrup/juice											
BL	85	93.4	65	86.7	64	86.5	57	85.1	62	87.3 **	0.197
PL	66	94.3	65	89.0	69	90.8	62	92.5	67	100 **	0.123
Nuts—salted											
BL	33	73.3	59	80.8	52	81.3	57	83.8	56	76.7	0.731
PL	22	78.6	24	77.4	24	82.8	30	90.9	27	90.0	0.083
Vegetables—canned											
BL	173	96.1	188	96.9	193	97.0	182	94.8	176	98.9	0.173
PL	68	98.6	72	98.6	75	98.7	52	100	58	96.7	0.606
Vegetables—pickled											
BL	53	40.8	48	33.8	79	45.1	77	42.1	63	37.5	0.913
PL	6	33.3	7	33.3	3	18.8	6	22.2	7	24.1	0.344
Salamis, hams, bacon											
BL	20	7.0	16	6.1	23	8.2	25	8.4	28	10.7	0.066
PL	1	2.6	1	2.5	1	2.5	2	4.9	1	2.3	0.858
Sausages, hotdogs											
BL	12	10.3	11	8.3	11	8.0	3	2.8	6	5.9	0.063
PL	0	0	3	6.5	0	0	0	0	1	5.0	0.982
Raw or frozen meats with flavour/coated											
BL	64	49.2	96	57.1	85	53.1	101	58.7	84	54.9	0.358
PL	30	75.0	27	69.2	14	58.3	20	58.8	23	62.2	0.138
Mayonnaise and salad dressings											
BL	1	0.7	2	1.4	0	0	0	0	0	0	0.086
PL	0	0	0	0	0	0	0	0	0	0	‡
Pasta sauces											
BL	95	63.3	96	64.0	106	63.1	91	65.0	99	65.6	0.661
PL	13	61.9	19	63.3	15	55.6	16	51.6	19	48.7	0.177
Savoury spreads and dips											
BL	47	20.2	60	22.2	86	30.6	96	30.1	95	31.6 **	<0.001
PL	20	52.6	19	47.5	15	51.7	16	51.6	18	56.3 **	0.679
Peanut butter and other nut-based spreads											
BL	43	91.5	57	89.1	58	93.5	68	89.5	74	92.5	0.792
PL	9	56.3	20	83.3	20	90.9	20	90.9	25	100	<0.001
Crisps and salty snacks											
BL	1	0.5	7	3.5	6	2.9	12	5.5	15	7.0	0.001
PL	0	0	2	4.3	2	4.7	2	5.7	3	8.6	0.087
Overall (all products)											
BL	1283	33.6	1377	33.7	1462	35.2	1501	34.5	1522	38.5 ***	0.082
PL	404	43.2	448	43.2	419	46.5	362	43.5	380	48.9 ***	0.342

HSR: Health Star Rating: BL: branded label; PL: private label. ^†^
*p*-values of chi-square tests for linear trend (linear-by-linear associations using Mantel–Haenzel tests). Comparisons of changes in proportions within private and branded labels in the period 2015–2019. ^‡^: Zero products with estimated HSR ≥ 3.5. * Pearson chi-square tests: *p* < 0.05; ** *p* < 0.005; *** *p* < 0.001. Comparisons of proportions between private and branded labels in 2019. Missing for estimated HSR-2015–2019 (n): savoury biscuits (19); sweet biscuits (101); everyday sliced breads (20); other breads (38); cakes/muffins: ready-to-eat (28); breakfast cereals: ready-to-eat (19); cereal bars (13); cheese: everyday cheeses (21); ice-cream (17); canned fish (38); fruit—canned in syrup/juice (13); nuts—salted (13); vegetables—canned (37); vegetables—pickled (67); processed meats-I (26); processed meats-II (18); processed meats-III (15); mayonnaise and salad dressings (56); pasta sauces (19); spreads I—savoury (27); spreads II—peanut butter and other nut-based spreads (4); crisps and snacks (18).

**Table 3 nutrients-13-02731-t003:** Number and proportion of branded and private label products displaying HSR on the pack by year (2015–2019), food category and overall.

Food Categories	HSR Displayed on Pack										*p* for Trend ^†^ Changes in Proportions In the Period 2015–2019
	2015		2016		2017		2018		2019		
	n	%	n	%	n	%	n	%	n	%	
Savoury biscuits											
BL	0	0.0	7	2.8	28	11.0	32	11.4	49	16.6 ***	<0.001
PL	0	0.0	26	49.1	34	55.7	45	71.4	40	80.0 ***	<0.001
Sweet biscuits											
BL	0	0.0	1	0.3	12	3.8	23	6.1	34	9.6 ***	<0.001
PL	0	0.0	18	20.2	33	50.8	56	90.3	66	90.4 ***	<0.001
Everyday sliced breads											
BL	0	0.0	0	0.0	0	0.0	2	1.8	16	15.2 ***	<0.001
PL	0	0.0	6	16.7	6	40.0	8	50.0	9	100.0 ***	<0.001
Other breads											
BL	0	0.0	2	0.9	8	4.0	11	5.0	22	9.7 ***	<0.001
PL	0	0.0	4	5.3	5	10.8	27	73.0	30	96.8 ***	<0.001
Cakes/Muffins: ready-to-eat											
BL	0	0.0	0	0.0	1	1.2	0	0.0	1	1.0 ***	0.360
PL	0	0.0	0	0.0	2	20.0	6	42.9	13	86.7 ***	<0.001
Breakfast cereals: ready-to-eat											
BL	5	2.3	106	39.8	119	48.2	136	52.7	156	55.7 ***	<0.001
PL	0	0.0	15	27.8	21	45.7	25	71.4	38	92.7 ***	<0.001
Cereal bars											
BL	0	0.0	14	7.6	34	18.8	39	21.7	55	35.3	<0.001
PL	1	2.8	13	28.9	14	35.0	13	37.1	14	41.2	<0.001
Ice-cream											
BL	0	0.0	1	0.3	4	1.4	5	1.6	11	3.2 ***	<0.001
PL	0	0.0	9	30.0	22	64.7	19	70.4	24	96.0 ***	<0.001
Canned fish											
BL	0	0.0	0	0.0	7	4.7	17	11.5	32	21.5 ***	<0.001
PL	0	0.0	20	25.0	34	44.2	39	78.0	40	95.2 ***	<0.001
Fruit—canned in syrup/juice											
BL	0	0.0	0	0.0	2	2.7	6	9.0	5	6.9 ***	<0.001
PL	0	0.0	14	19.2	25	32.9	42	62.7	62	89.9 ***	<0.001
Nuts—salted											
BL	0	0.0	7	9.1	26	40.0	26	38.2	24	32.0 ***	<0.001
PL	0	0.0	5	16.1	20	69.0	28	84.8	26	83.9 ***	<0.001
Vegetables—canned											
BL	3	1.5	6	4.5	24	11.9	37	19.3	44	24.2 ***	<0.001
PL	0	0.0	13	17.3	23	30.3	34	65.4	54	90.0 ***	<0.001
Vegetables—pickled											
BL	0	0.0	0	0.0	13	7.1	20	10.4	26	14.7 ***	<0.001
PL	0	0.0	1	4.5	2	11.8	14	50.0	18	60.0 ***	<0.001
Processed meats I: salamis, hams, bacon											
BL	0	0.0	0	0.0	17	6.0	18	6.1	17	6.5 ***	<0.001
PL	1	2.6	10	24.4	21	52.5	29	70.7	39	88.6 ***	<0.001
Processed meats II: sausages, hotdogs											
BL	0	0.0	0	0.0	3	2.2	11	10.2	6	5.8 ***	<0.001
PL	0	0.0	1	2.2	7	29.2	9	33.3	13	65.0 ***	<0.001
Processed meats III: raw or frozen meats with flavour/coated											
BL	0	0.0	1	0.6	46	28.4	51	29.7	46	30.1 ***	<0.001
PL	0	0.0	2	5.0	17	70.8	25	73.5	27	71.1 ***	<0.001
Mayonnaise and salad dressings											
BL	0	0.0	0	0.0	1	0.5	2	1.1	2	1.3 ***	0.054
PL	0	0.0	5	21.7	6	28.6	22	84.6	33	97.1 ***	<0.001
Pasta sauces											
BL	0	0.0	13	8.6	29	17.3	32	22.5	56	35.9 ***	<0.001
PL	0	0.0	9	30.0	15	55.6	21	67.7	36	92.3 ***	<0.001
Spreads I: savoury spreads and dips											
BL	0	0.0	12	4.4	33	11.7	35	11.0	39	12.7 ***	<0.001
PL	0	0.0	5	12.5	12	41.4	16	51.6	32	100.0 ***	<0.001
Spreads II: peanut butter and other nut-based spreads											
BL	0	0.0	14	21.9	19	30.2	22	28.6	21	26.3 ***	0.002
PL	0	0.0	12	50.0	19	86.4	19	86.4	25	100.0 ***	<0.001
Crisps and salty snacks											
BL	0	0.0	1	0.5	5	2.4	10	4.6	14	6.5 ***	<0.001
PL	0	0.0	2	4.3	27	62.8	30	85.7	34	97.1 ***	<0.001
Overall (all products)											
BL	08	0.2	188	4.4	431	10.2	540	12.2	681	17.2 ***	<0.001
PL	02	0.2	195	18.5	398	44.1	569	68.2	718	92.4 ***	<0.001

HSR: Health Star Rating: BL: branded label; PL: private label. ^†^
*p*-values of chi-square tests for linear trend (linear-by-linear associations using Mantel–Haenzel tests). Comparisons of changes in proportions within private and branded labels in the period 2015–2019. *** *p* < 0.001. Comparisons of proportions between private and branded labels in 2019.

**Table 4 nutrients-13-02731-t004:** Mean (SD) price (in New Zealand dollars) between branded label and private label products by food category and overall, 2019.

Food Categories	Mean Price (NZ$/100 g)		*p* ^†^
	n	Mean (SD)	
Savoury biscuits			<0.001
BL	274	2.33 (1.24)	
PL	50	1.51 (1.37)	
Sweet biscuits			<0.001
BL	306	2.07 (1.76)	
PL	72	1.06 (0.84)	
Everyday sliced breads			^‡^
BL	92	0.66 (0.42)	
PL	9	0.57 (0.52)	
Other breads			<0.001
BL	210	1.30 (0.61)	
PL	30	0.82 (0.26)	
Cakes/Muffins: ready-to-eat			^‡^
BL	83	1.84 (0.75)	
PL	15	1.53 (0.66)	
Breakfast cereals: ready-to-eat			<0.001
BL	244	1.59 (0.96)	
PL	40	0.84 (0.38)	
Cereal bars			<0.001
BL	139	2.07 (1.04)	
PL	33	1.15 (0.09)	
Ice-cream			^‡^
BL	299	1.59 (1.33)	
PL	25	0.66 (0.42)	
Canned fish			<0.001
BL	140	1.94 (0.82)	
PL	42	1.27 (0.44)	
Fruit—canned in syrup/juice			0.009
BL	68	0.50 (0.14)	
PL	68	0.41 (0.22)	
Nuts—salted			^‡^
BL	61	3.57 (3.07)	
PL	29	2.11 (0.95)	
Vegetables—canned			<0.001
BL	163	0.53 (0.20)	
PL	59	0.31 (0.14)	
Vegetables—pickled			0.068
BL	142	1.94 (1.38)	
PL	29	1.45 (0.92)	
Processed meats I: salamis, hams, bacon			0.178
BL	236	3.07 (2.36)	
PL	39	2.54 (1.76)	
Processed meats II: sausages, hotdogs			^‡^
BL	87	1.74 (0.97)	
PL	9	0.86 (0.43)	
Processed meats III: raw or frozen meats with flavour/coated			0.658
BL	141	1.69 (0.71)	
PL	30	1.76 (0.92)	
Mayonnaise and salad dressing			0.052
BL	127	1.66 (0.97)	
PL	32	1.29 (0.84)	
Pasta sauces			0.756
BL	124	1.24 (1.31)	
PL	39	1.31 (1.17)	
Spreads I: savoury spreads and dips			<0.001
BL	236	1.91 (1.00)	
PL	32	1.15 (0.59)	
Spreads II: peanut butter and other nut-based spreads			0.018
BL	65	2.25 (1.59)	
PL	24	1.40 (1.15)	
Crisps and salty snacks			<0.001
BL	203	2.10 (1.12)	
PL	35	1.23 (0.32)	
Overall (all products)			<0.001
BL	3440	1.83 (1.44)	
PL	741	1.16 (0.98)	

NZ$: New Zealand dollars; SD: standard deviation; BL: branded label; PL: private label. ^†^ Student *t*-tests for comparison of means of two independent samples. ^‡^: Comparisons between means were not performed when PL had n < 30 products.

## Data Availability

Because of the commercial and legal restrictions to the use of copyrighted material, it is not possible to share data openly, but unredacted versions of the dataset are available with a licensed agreement that they will be restricted to non-commercial use. For access to Nutritrack, please contact the National Institute for Health Innovation at the University of Auckland at enquiries@nihi.auckland.ac.nz.

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
