# Peer review of "Comparison of Healthiness, Labelling, and Price between Private and Branded Label Packaged Foods in New Zealand (2015–2019)"

_nutrients, 2021, doi:10.3390/nu13082731_

Round 1

Reviewer 1 Report

I would like to start by congratulating the authors for the work they have done. However, I consider that its publication is not possible since I consider that there are elements that would need to be modified and that I will forward to you below.

Theoretical background

  • The work lacks a theoretical foundation, since the elements dealt with are not delimited. I consider it necessary to specify the different theoretical elements dealt with in the work.
  • In the work the privated label (PL) is contrasted with the branded label (BL). The privated label can bring together a wide range of specific types of brands that can be perceived, even with a higher quality than the branded labels.

Method

  • Healthiness (through sodium and sugar content) and price are analyzed in the article. Why haven't you focused exclusively on healthiness? To this must be added the different time horizon used to mediate both variables, 2015/2019 and 2018/2019.
  • In the Introduction it indicates that the supermarket environment is a duopoly in New Zealand, however, in the section where it presents the “healthiness” it indicates that there are four. To this is added that it mentions other types of establishments on line 115.
  • Why do you say that supermarkets seek to generate consumer welfare through lower prices? Does this trend condition your results relative to the price analysis?
  • If the percentage of the privated label (PL) on sales is 5.0% and 5.2% in the main supermarkets, are the results still relevant as part of the work is focused on the study of PL.
  • While in the Abstract they indicate that they compared those products that "display the voluntary HSR nutrition label" (line 16), later (line 98) they indicate that "we estimated the HSR score for all products ..."

Analysis

  • In the introduction it indicates that it uses the chi-square test to detect trends. I believe that this statement requires more precision.
  • In the statistical analysis, it is not clear that the sodium or sugar content can be the dependent variable, as well as the coding of the year as an independent variable (2015 = 0, 2016 = 0.25, 2017 = 0.50, 2018 = 0.75, 2019 = 1). References should be presented that support the use of this technique, selection of variables and coding.

Style

  • There is some typo, such as “Changee”.

Author Response

Manuscript ID: nutrients-1317391

Manuscript: Comparison of healthiness, labelling, and price between private and
branded label packaged foods in New Zealand (2015-19)

To the Editor of Nutrients and the reviewers of the manuscript,

Thank you for your feedback on this piece of work and for the suggestions provided, which improved this revised version of the manuscript. This letter outlines the response to each point raised by the reviewers and indicates where in the text the amendments can be found.  According to the journal`s request, all changes made in the revised version of the mansucript used “track changes” function of MS Word.

Sincerely,Dr Teresa Gontijo de Castro

Senior Research Fellow

University of Auckland

REVIEWER 1

Open Review

English language and style

( ) Extensive editing of English language and style required
( ) Moderate English changes required
(x) English language and style are fine/minor spell check required
( ) I don't feel qualified to judge about the English language and style

Response: Thanks for this careful revision. We have checked the revised version of the manuscript to fix any spelling mistakes/errors.

Yes

Can be improved

Must be improved

Not applicable

Does the introduction provide sufficient background and include all relevant references?

( )

( )

(x)

( )

Is the research design appropriate?

( )

( )

(x)

( )

Are the methods adequately described?

( )

( )

(x)

( )

Are the results clearly presented?

( )

(x)

( )

( )

Are the conclusions supported by the results?

( )

(x)

( )

( )

Comments and Suggestions for Authors

I would like to start by congratulating the authors for the work they have done. However, I consider that its publication is not possible since I consider that there are elements that would need to be modified and that I will forward to you below.

Response: Thank you for the suggestions provided. We clarified below some of the questions/issues raised by the reviewer and when applicable, we made the suggested changes in the revised version of the manuscript.

Theoretical background

  • The work lacks a theoretical foundation, since the elements dealt with are not delimited. I consider it necessary to specify the different theoretical elements dealt with in the work.
  • In the work the privated label (PL) is contrasted with the branded label (BL). The privated label can bring together a wide range of specific types of brands that can be perceived, even with a higher quality than the branded labels.

Response: We are more than happy to provide more details on the theoretical foundation required by the reviewer. However, we are not sure of what exactly would be the theoretical elements the reviewer suggests need to be specified. Could you please clarify this point for us? Thus, we can incorporate it into the final version of the manuscript.

Method

  • Healthiness (through sodium and sugar content) and price are analyzed in the article. Why haven't you focused exclusively on healthiness? To this must be added the different time horizon used to mediate both variables, 2015/2019 and 2018/2019.

Response: We also agree with the reviewer that this paper has enough findings to limit the comparisons to healthiness of supermarket private label and branded label foods. However, we also wanted to “draw a complete picture” when making these comparisons by also comparing how “transparent” PL and BL brands are when it comes to inform the consumers how healthy or unhealthy the products they are purchasing are. We also wanted to explore if there were differences in prices between PL and BL options. For consumers it is very relevant to know if healthier PL options are cheaper, more expensive or similar to BL options as price is a major influence on food choice.

In relation to the next point raised about the years we included in the analyses, we believe our objectives are clear and justify the use of the data of 2019 and the use of data of 2015,2016,2017,2018, and 2019. In this study we aimed:

  1. to describe and compare in 2019 supermarket PL and of BL food products in relation to their healthiness, proportion displaying HSR in the package and price and;
  2. to examine if there were changes over time (across 2015-2019) in products` healthiness and proportion of products displaying HSR among PL and among BL products.

In the Introduction it indicates that the supermarket environment is a duopoly in New Zealand, however, in the section where it presents the “healthiness” it indicates that there are four. To this is added that it mentions other types of establishments on line 115.

Response: Thank you for this important point raised as this aspect was confusing in our paper. We have reviewed the paper aiming to clarify the point raised by the reviewer. The NZ duopoly refers to two supermarket retailers, which provide food items to the 8 NZ supermarket chains. Amongst these supermarket chains, four of them represent major chains in terms of volume sales (New World, Four Square, Countdown and PAK’nSAVE). Please consult figure below which illustrates the NZ supermarket retailers and the supermarket`s chains covered by each retailer.

Source: https://www.stuff.co.nz/business/300368920/government-responds-to-damning-supermarket-report-on-high-costs

We have made further changes in the manuscript to clarify the point raised by the reviewer:

Introduction (page 2):

“In NZ, supermarkets account for ~75% of all purchases of packaged foods [8], and the supermarket food environment consists of a duopoly of two grocery retailers: Foodstuffs and Woolworths. These two supermarket retailers provide groceries to eight supermarket chains across the country [9].”

“Both supermarket retailers in NZ offer PL options, also known as “own-brands”, “generic brands”, “store brands”, or “economy lines”.

“Since 2016, both NZ supermarket retailers have made commitments to improve the healthiness of their PL products, specifically through reformulation focused on reducing sugar, sodium, and saturated fat content [12, 13, 14]. For example, one retailer has committed to all PL products being nutritionally on par, or better than, the average comparable BL product (by 2018) [12]. Both retailers have also committed to displaying the voluntary Health Star Rating (HSR) nutrition label on all applicable PL products [12, 13, 14].”

Discussion (pages 24 & 25; lines 124-138)

“In our study we did not evaluate how each of the two supermarket retailers met their commitments made in 2016 because for most food categories assessed there were insufficient PL products to provide robust comparisons.”

“Both supermarket retailers committed to improving the nutrition of their PL products.”

“Together, these findings indicate that the commitments of supermarket retailers have been partially met, but more work is needed to increase the proportion of products with HSR >3.5 across all types of foods.”

Why do you say that supermarkets seek to generate consumer welfare through lower prices? Does this trend condition your results relative to the price analysis?

Response: We have reworded this part of the introduction to better reflect the idea we wanted to display here.

Introduction (page 2; lines 52-53):

“The availability of private label (PL) and branded label (BL) products in supermarkets are important for generating price competitiveness and to offer consumers options of products` quality and variety [17].”

If the percentage of the privated label (PL) on sales is 5.0% and 5.2% in the main supermarkets, are the results still relevant as part of the work is focused on the study of PL.

Response: Yes, these results are important, as, overall, in 2020, PL sales accounted for 10.2% of packaged foods sold in NZ stores and on-line. We have changed this part of the introduction presenting the aggregated figure rather than separated by the two NZ supermarket retailers. This information was taken from Euromonitor passport.

Introduction (page 2; line 55):

“In 2020, PL sales accounted for 10.2% of packaged foods sold in NZ stores and online [8].”

While in the Abstract they indicate that they compared those products that "display the voluntary HSR nutrition label" (line 16), later (line 98) they indicate that "we estimated the HSR score for all products ..."

 Response: We understand the confusion here.  However, it is important to clarify that these are two distinct indicators that were compared between PL and BL products.

The first indicator compared the proportion of PL and BL products which had estimated HSR >3.5 (this is an indicator of healthiness). We (researchers) estimated HSR for all products based on the information contained in products` NIPs (nutrition information panels). We needed to estimate HSR of products because, as we explained in the methods section of the manuscript, “by 2018, only 21% of the NZ supermarket packaged products displayed the manufacturer calculated HSR score on the pack’. We chose to estimate HSR for all products included in the analyses (products that displayed and products that did not display the HSR) to ensure the same estimation method was used for all products.

The second indicator compared the proportion of PL and BL products which displayed HSR information in their packages. This is an indicator of “transparency”, as we wanted to examine if manufacturers are informing consumers about how healthy or unhealthy their products are.

We believe that the abstract and the manuscript`s methods are clear regarding the point raised by the reviewer. However, if the reviewer would like to suggest wording that he(she) believes clarifies the point raised, we can incorporate the suggested changes in the text.

Analysis

In the introduction it indicates that it uses the chi-square test to detect trends. I believe that this statement requires more precision.

Response: We believe the reviewer is referring to the abstract section. We acknowledged that we could have been more specific in relation to the test performed in the abstract. As described in the methods section of the manuscript, the five-year trends in the proportion of PL and BL products with HSR >3.5  and products displaying the HSR were examined using chi-square tests for trends (linear-by-linear associations using Mantel Haenzel tests). We have added more information about this test in the revised version of the abstract.

In the statistical analysis, it is not clear that the sodium or sugar content can be the dependent variable, as well as the coding of the year as an independent variable (2015 = 0, 2016 = 0.25, 2017 = 0.50, 2018 = 0.75, 2019 = 1). References should be presented that support the use of this technique, selection of variables and coding.

Response:  To clarify the points raised, we have re-written this part of the statistical methods section as below:

Methods- Statistical analyses (page 6; lines 163-166):

“To estimate the average change in sodium (mg/100g) or sugar (g/100g) contents from 2015 to 2019, linear regression models were performed with sodium or sugar as the dependent variable. Year was included in the model as the independent variable-as a continuous variablecoded as: 2015=0, 2016=0.25, 2017=0.50, 2018=0.75, and 2019=1 [similar to analyses conducted elsewhere-3, 26].”

We have also added two recent references that have used the same analytical approach we used in the present manuscript to estimate the average change in products` nutrient content across the years. This analytical approach was suggested by two experienced senior biostatisticians who are also authors in the referred papers below (Dr. Y Jiang  and Professor T Blakeley). In the present paper and in the papers below we were interested in estimating the average change in nutrient contents of products across certain number of years, rather than comparing changes in means between two years.

  1. Gontijo de Castro, T.; Eyles, H.; Ni Mhurchu, C.; Young, L.; Mackay, S. Seven-year trends in the availability, sugar content and serve size of single-serve non-alcoholic beverages in New Zealand: 2013–2019. Public Health Nutr. 2021, 24, 1595-1607, doi:10.1017/S1368980020005030
  2. Eyles, H., Jiang, Y., Blakely, T. et al. Five-year trends in the serve size, energy, and sodium contents of New Zealand fast foods: 2012 to 2016. Nutr J 17, 65 (2018). https://doi.org/10.1186/s12937-018-0373-7

Style

There is some typo, such as “Changee”.

Response: Thank you for this careful revision. We have checked the revised version of the manuscript to fix any spelling mistakes and typos.

Reviewer 2 Report

The article portrays a current topic that is very pertinent and of great interest to public health.

The article is well-structured, well-founded, has a clear objective, adequate methodology, well-selected results to respond to the objective of the study given the methodology used. The interesting discussion could benefit from the integration of international studies that support the conclusions they present regarding the importance of the topic, future recommendations for health policies. Thus, the inclusion of articles from the MDPI group is suggested, such as:

Feteira-Santos, R.; Alarcão, V.; Santos, O.; Virgolino, A.; Fernandes, J.; Vieira, C.P.; João Gregório, M.; Nogueira, P.; Costa, A.; Graça, P. Looking Ahead: Health Impact Assessment of Front-of-Pack Nutrition Labelling Schema as a Public Health Measure. Int. J. Environ. Res. Public Health 2021, 18, 1422. https://doi.org/10.3390/ijerph18041422

In the section on the discussion, in addition to the aforementioned recommendations, it is specifically suggested scientific support regarding “Findings in relation to the commitments made by NZ supermarkets”.

Some sentences (line 124-128) need more discussion and scientific support, such as “Together, these findings indicate that the commitments of supermarkets have been partially met, but more work is needed to increase the proportion of products with HSR >3.5 across all types of foods”

The same applies to “There are no public commitments made by the NZ supermarket retailers on price of PL products, probably because PL products are usually lower in price than branded ones, and price is generally considered commercially sensitive.”

Also, generally in the “Implications of the findings” section (line 148-157) more discussion and scientific studies could be added according to front of package nutrition labelling experiences in different countries.

The figures presented contain very relevant information but need to be redone as they appear unformatted.

Author Response

REVIEWER 2

Open Review

English language and style

( ) Extensive editing of English language and style required
( ) Moderate English changes required
(x) English language and style are fine/minor spell check required
( ) I don't feel qualified to judge about the English language and style

Response: Thanks for this careful revision. We have checked the revised version of the manuscript to fix any spelling mistakes/errors.

Yes

Can be improved

Must be improved

Not applicable

Does the introduction provide sufficient background and include all relevant references?

(x)

( )

( )

( )

Is the research design appropriate?

(x)

( )

( )

( )

Are the methods adequately described?

(x)

( )

( )

( )

Are the results clearly presented?

(x)

( )

( )

( )

Are the conclusions supported by the results?

(x)

( )

( )

( )

Comments and Suggestions for Authors

The article portrays a current topic that is very pertinent and of great interest to public health.

The article is well-structured, well-founded, has a clear objective, adequate methodology, well-selected results to respond to the objective of the study given the methodology used. The interesting discussion could benefit from the integration of international studies that support the conclusions they present regarding the importance of the topic, future recommendations for health policies. Thus, the inclusion of articles from the MDPI group is suggested, such as:

Feteira-Santos, R.; Alarcão, V.; Santos, O.; Virgolino, A.; Fernandes, J.; Vieira, C.P.; João Gregório, M.; Nogueira, P.; Costa, A.; Graça, P. Looking Ahead: Health Impact Assessment of Front-of-Pack Nutrition Labelling Schema as a Public Health Measure. Int. J. Environ. Res. Public Health 2021, 18, 1422. https://doi.org/10.3390/ijerph18041422

Response: Thank you very much for this positive feedback and for clearly understanding the importance and implications of the study. As suggested, we have added some discussion (and references) about the impact of mandatory front-of pack labelling for: improving consumer`s awareness of products` healthiness to provide them choices and; to stimulate reformulation of products and improvement of their healthiness.

Discussion (Pages 22 lines 100-108)

“Front of pack labelling provides visual information on product nutritional contents and studies have shown that it influences consumer`s knowledge [34,35] and products reformulation [36]. Recent systematic review and meta-analyses including controlled experimental/intervention and interrupted time series found that findings about influence of front of pack labelling on consumers` consumption were limited and inconsistent. However, evidence from experimental and ‘real-life’ studies shows that front of pack labelling encouraged healthier purchasing [34]. An on-line randomized-controlled study of a large representative British sample found that front of pack labelling improved participants` ability to correctly rank products according to their healthiness [35]. A non-experimental prospective study reported that food reformulation occurred after the first phase of the Chilean Food Labelling and Advertising Law, with significant decreases in the amount of sugars and sodium in several groups of packaged foods and beverages between 2015 and 2017 [36].”

In the section on the discussion, in addition to the aforementioned recommendations, it is specifically suggested scientific support regarding “Findings in relation to the commitments made by NZ supermarkets”.

Response: We are sorry we were not able to respond to this suggestion as it is not clear to us what scientific support means in this context. In our paper we discussed our findings from our robust scientific methodology to evaluate if the commitments made by the NZ supermarket retailers were met. Our conclusions in this space were based on our findings and also taking into consideration the limitations of our data. For example, in our study we did not evaluate how each of the two supermarket retailers met their commitments made in 2016, because for most food categories assessed there were insufficient PL products to provide robust comparisons.

Some sentences (line 124-128) need more discussion and scientific support, such as “Together, these findings indicate that the commitments of supermarkets have been partially met, but more work is needed to increase the proportion of products with HSR >3.5 across all types of foods”

Response: We are sorry we were not able to respond to this suggestion as it is not clear to us what scientific support means in this context. In our paper we discussed our findings (from our robust scientific methodology) to evaluate if the commitments made by the NZ supermarket retailers were met. Our conclusions in this space were based on our findings and taking into consideration the limitations of our data. For example, in our study we did not evaluate how each of the two supermarket retailers met their commitments made in 2016, because for most food categories assessed there were insufficient PL products to provide robust comparisons.

The same applies to “There are no public commitments made by the NZ supermarket retailers on price of PL products, probably because PL products are usually lower in price than branded ones, and price is generally considered commercially sensitive.”

Response: We have reworded this part of the discussion, to make clearer that this is our interpretation of why we believe supermarket retailers have not made commitments regarding price of PL products.

Discussion (Page 25;  lines 139-140)

“There are no public commitments made by the NZ supermarket retailers on price of PL products. We believe that this is probably because PL products are usually considered lower cost options than branded ones, and price is generally considered commercially sensitive.”

Also, generally in the “Implications of the findings” section (line 148-157) more discussion and scientific studies could be added according to front of package nutrition labelling experiences in different countries.

Response: As suggested, we have added some discussion (and references) about the impact of mandatory front-of pack labelling for: improving consumer`s awareness of products` healthiness-providing them choices and; to stimulate reformulation of products and improvement of their healthiness.

Discussion (Page 22; lines 100-108)

“Front of pack labelling provides visual information on product nutritional contents and studies have shown that it influences consumer`s knowledge [34,35] and products reformulation [36]. Recent systematic review and meta-analyses including controlled experimental/intervention and interrupted time series found that findings about influence of front of pack labelling on consumers` consumption were limited and inconsistent. However, evidence from experimental and ‘real-life’ studies shows that front of pack labelling encouraged healthier purchasing [34]. An on-line randomized-controlled study of a large representative British sample found that front of pack labelling improved participants` ability to correctly rank products according to their healthiness [35]. A non-experimental prospective study reported that food reformulation occurred after the first phase of the Chilean Food Labelling and Advertising Law, with significant decreases in the amount of sugars and sodium in several groups of packaged foods and beverages between 2015 and 2017 [36].”

The figures presented contain very relevant information but need to be redone as they appear unformatted.

Response: Figures 1 and 2 were formatted.

Round 2

Reviewer 1 Report

I would like to congratulate the authors for the work they have done in responding so quickly to the suggested modifications. Now, I consider that the work is suitable for publication although I recommend reviewing certain minor elements that I indicate below:

  • In line 150 appears what might appear to be a misprint "categories (s)"
  • I would recommend indicating that "Table S1" and "Table S2" are information available in the "Supplementary files"
  • Line 192 "conducted elsewhere-3, 26]"
  • I don't understand why the paragraph between lines 113 (page 19) and 124 (page 20) is in quotation marks

Author Response

REVIEWER 1 

Open Review

English language and style

( ) Extensive editing of English language and style required
( ) Moderate English changes required
(x) English language and style are fine/minor spell check required
( ) I don't feel qualified to judge about the English language and style

Yes

Can be improved

Must be improved

Not applicable

Does the introduction provide sufficient background and include all relevant references?

(x)

( )

( )

( )

Is the research design appropriate?

(x)

( )

( )

( )

Are the methods adequately described?

(x)

( )

( )

( )

Are the results clearly presented?

(x)

( )

( )

( )

Are the conclusions supported by the results?

(x)

( )

( )

( )

Comments and Suggestions for Authors

I would like to congratulate the authors for the work they have done in responding so quickly to the suggested modifications. Now, I consider that the work is suitable for publication although I recommend reviewing certain minor elements that I indicate below:

Response: Thank you very much for your prompt review, for identifying the mistakes below and, also for the suggestions provided. We have included your suggestions and fixed all the mistakes you identified in this version of the manuscript. Please, note that in this second review of the manuscript we reworded part of the methods section that describes the nature of the data collected by Nielsen Homescan (page 3, lines 117-123).

We have used word tracked changes for easy identification of the changes made in this version of the manuscript.

In line 150 appears what might appear to be a misprint "categories (s)"

Response: Thank you. This was fixed.

I would recommend indicating that "Table S1" and "Table S2" are information available in the "Supplementary files"

Response: Thank you. We added this ‘(supplementary file)” when we referred to tables S1, S2 and also when we referred to tables S3 and S4, for consistency.

Line 192 "conducted elsewhere-3, 26]"

Response: Thank you. This was fixed.

I don't understand why the paragraph between lines 113 (page 19) and 124 (page 20) is in quotation marks

Response: Thank you. This was a typing error. We have removed the quotation marks.